# Data Acquisition for Condition Monitoring in Tactical Vehicles: On-Board Computer Development

**DOI:** 10.3390/s23125645

**Published:** 2023-06-16

**Authors:** Francisco Jose Ochando, Antonio Cantero, Juan Ignacio Guerrero, Carlos León

**Affiliations:** 1Emerging Technologies Center in the Spanish Army, Av. Radio Televisión S/N, 28223 Pozuelo de Alarcon, Spain; 2Higher Polytechnic School of the Army, C/Joaquin Costa, 6, 28002 Madrid, Spain; acanobr@oc.mde.es; 3Department of Electronic Technology, High Polytechnic University School, University of Seville, C/Virgen de África, 7, 41011 Seville, Spain; juaguealo@us.es (J.I.G.); cleon@us.es (C.L.); 4Polytechnic University School of Computing Engineering, University of Seville, Av. Reina Mercedes S/N, 41012 Seville, Spain

**Keywords:** data acquisition, military vehicles, fault diagnosis, generic vehicle architecture, condition monitoring, interoperability

## Abstract

This paper outlines the development of an onboard computer prototype for data registration, storage, transformation, and analysis. The system is intended for health and use monitoring systems in military tactical vehicles according to the North Atlantic Treaty Organization Standard Agreement for designing vehicle systems using an open architecture. The processor includes a data processing pipeline with three main modules. The first module captures the data received from sensor sources and vehicle network buses, performs a data fusion, and saves the data in a local database or sends them to a remote system for further analysis and fleet management. The second module provides filtering, translation, and interpretation for fault detection; this module will be completed in the future with a condition analysis module. The third module is a communication module for web serving data and data distribution systems according to the standards for interoperability. This development will allow us to analyze the driving performance for efficiency, which helps us to know the vehicle’s condition; the development will also help us deliver information for better tactical decisions in mission systems. This development has been implemented using open software, allowing us to measure the amount of data registered and filter only the relevant data for mission systems, which avoids communication bottlenecks. The on-board pre-analysis will help to conduct condition-based maintenance approaches and fault forecasting using the on-board uploaded fault models, which are trained off-board using the collected data.

## 1. Introduction

The most valuable goal of logistics in the army is to maintain the impulse of operations by achieving the highest operability. Another objective is to maintain a cost-effective and safety-assured maintenance strategy [1]. The first objective is achieved by quick response times, avoiding minimum product damage, and maximizing operation by using the maximized mean time between failures (MTBF). The second objective is related to ensuring the well-driven, efficient fluency of logistic and maintenance operations, minimizing the mean time to repair (MTTR) and the economization of resources. When performing operations, a quick maintenance response is desirable; however, due to the unpredictability of faults, it can be very complex and challenging to obtain an optimal response in the logistic flow. Traditionally, maintenance time and extent have been determined by two factors: the technical condition of the vehicle and a regular maintenance schedule [2]. This maintenance approach is mainly based on a preventive maintenance program for avoiding failures and corrective maintenance to fix current faults. This means that maintenance is triggered according to the presence of a failure or a statistically calculated mean time to lost performance according to the experience or knowledge of the manufacturer. Preventive vehicle maintenance is scheduled at a fixed time or mileage, assuming that some degradation or reduced performance could appear. For the reasons mentioned above, the use of a corrective preventive maintenance scheme could not be cost-effective and efficient, especially when compared with low-mileage vehicles or efficiently driven vehicles; consequently, some parts of the equipment can be replaced before the risk of failure has been reached [3]. In contrast, tactical vehicles that operate in harsh environmental conditions can be affected by premature failures, which frequently occur before the theoretically prescribed mileage.

The main approach for achieving a faster and more efficient maintenance is to implement a data-driven maintenance approach. To achieve this new approach, more frequent inspections must be conducted, which is an expensive method in terms of the time and people required. However, the best and fastest method for learning the status of the system is by continuously registering functional data for further analyses. We are aware of the following problems:Tactical radio systems have limited bandwidth and unstable connections for sending data in real time in contrast to industrial fleets, which use commercial networks.The operating conditions of the on-board device can be extreme and situations of mechanical and environmental stress may occur, making it necessary to design the system in a resistant way and create rugged devices.The regulation and validation of on-board devices in tactical vehicles has many restrictions, which are not only physical restrictions but are also linked to military regulation.The volume of data needed for reliable data-driven maintenance, as well as the transmission of this much data is challenging.

The proposed solution was developed in two stages:The on-board device is in the acquisition stage: In this stage, the device only stores, translates, and exchanges vehicle data with the systems. In this case, the on-board device is optimized to store the information to be extracted in the next maintenance review.The on-board device is in operation mode: In this stage, the on-board device continues to store the information; however, it also stores the maintenance models that are created from the data extracted in the first stage, and only the alarms or event that are generated are transmitted thanks to the processing at the edge. This part of the development is not included in this paper.

For the above-mentioned reason, on-board edge computing is needed to treat and minimize the transmitted data ahead of time. The current paper provides the design and development of this on-board device, where the acquisition system is based on the CAN bus and the results of tests in real tactical vehicles are presented. This study will be a proof of concept of a computer system for registering and computing vehicle sensors and network data. Thus, the main contributions of this paper are as follows:The design and implementation of an on-board device in a tactical vehicle is presented.The design of the acquisition infrastructure is created as needed for health and use monitoring system (HUMS) integration and deployment.The proposed on-board device is in accordance with current military regulations and standards.The application is presented in a real case involving tactical vehicles deployed in conflict zones.

As far as we know, there are no other references of this type that offer details on the implementation and testing of computers for data acquisition and further analysis using tactical vehicles.

The next section presents the related work on commercial vehicles and tactical vehicle studies. Section 3 presents a brief introduction to condition and predictive maintenance, including a brief presentation on different maintenance analysis approaches. The next subsections include using the addressed methods to develop a complete edge computing platform for data acquisition and monitoring according to NATO standards on a generic open vehicle architecture. The system will be capable of storing data, condition monitoring by filtering problem information data, and limiting the transmission to mission systems into an open architecture for command, control, and logistics. Section 4 includes the testing procedures and data prospecting methods for communications efficiency. Finally, we evaluated the amount of data that are needed for on-board computer analysis and the resources to avoid using tactical communications for the massive transmission of data to an external analysis tool.

## 2. Literature Review

The next section presents the related work on commercial vehicles and tactical vehicle studies. Some work has been developed for different vehicle analysis purposes. As an example, a vehicle fault detection system proposed in [4] based on a machine learning model trained over offline car simulation data can detect a vehicle failure based on large and fast streaming telemetry data from vehicles. The work focused on data acquisition and communication from multiple vehicles to an off-board distributed system for data analysis. Other simple data-logging systems have been developed for remote data analysis using remote telemetry [5]. A data-driven method was used for abnormal battery charging for electrical vehicles in [6]. AUTOSAR [7] is a worldwide development partnership of vehicle manufacturers, for open system architecture development for health and use monitoring purposes according to the requirements of the ISO 26262 standard [8]. This project tries to implement functions such as life supervision, deadline supervision, logical supervision, and health status supervision. Life supervision involves periodically checking checkpoints for the correct functioning of systems, and deadline supervision checks the time span of transitions between two checkpoints of a supervised entity. Logical supervision is a technique to verify the correct execution of the system. The health state supervisor system can connect external and unbounded supervision results to the Health Monitoring. Based on the results of the supervision functions, the Health Monitoring tries to determine the appropriate actions. Some authors aimed to investigate the use of publish–subscribe protocols for transport telemetry data with a hierarchical structure of access to a set of vehicles and their parameters in the field of logistics [9].

The proliferation of sensors and the deployment of Internet of Things (IoT) devices in the military domain also requires the development of an IoT protocol stack that provides advantages for its use in the military Internet of Things (MIoT) domain. The use of standard commercial computer devices (off-the-shelf) for the implementation of MIoT networks in an NGVA scenario for the interoperability application between unmanned ground vehicles (UGVs) has been considered in [10]. The Land vehicle with Open System Architecture (LOSA) [11] or Land Vehicle Open System Architecture (LAVOSAR) was the first study based on high-level operational vision focused on tactical vehicles, as we can see in Figure 1. This project, initiated by the European Defense Agency (EDA), developed a roadmap to harmonize the procedures that will enable data exchange, including procedures in tactical vehicles. The LAVOSAR objectives included the creation of a Normative Framework, and the study and development of a functional and technical mission system architecture supporting an open architecture approach. The study identified technologies applicable to on-board systems, but are not properly identified and standardized. LAVOSAR II [12] addressed the identified gaps and tried to collaborate with stakeholders to establish a mature architecture based on standards to complement the NATO Generic Vehicle Architecture (NGVA) [13] and Def Stan 23-09 [14] to develop an embedded system for British military vehicles into a Generic Vehicle Architecture. The LAVOSAR II studies identified and defined an additional architectural layer in LAVOSAR, updating the Open Reference Architecture standards, logistic procedures, and operational workflows defined in LAVOSAR’s study. As a result, an Open Architecture Land Vehicle Architecture Model was developed as the NATO Standard Agreement (STANAG) for interoperability number 4754 [15] and a proposed verification and validation specification [16].

## 3. Materials and Methods

The Condition Base Maintenance (CBM) approach requires frequent data on vehicle health and use. Reliability-Centered Maintenance (RCM) is an approach which we are expecting to gain longer up-times, lower costs, and better control and decisions. This kind of maintenance should be consistent with the life expectancy or the prognostic value of the system failure. ISO 13381-1 defines “prognostics” as the estimate of the time to failure and the risk of one or more existing and future failure modes. Prognostic Health Maintenance (PHM) monitors the information on the condition of the product, detects fault symptoms, and predicts the trend of the fault and thus making the transition to predictive maintenance. PHM may contain model-based methods for the remaining useful life (RUL) [17] of the systems. The remaining useful life (RUL) is a metric recognized as a key feature, and the estimate of useful life allows for an advantage over faults or the avoidance of unnecessary maintenance costs [18]. The RUL can be determined using a theoretical Fault-Mode Effects Analysis (FMEA) or by calculating a prognostic model. A prognostic model uses a data-driven analysis using machine learning techniques, knowledge-based statistical rules, or neural networks validated from real-time registered vehicle sensor values and system diagnostics [19]. The final purpose of a PHM or CBM will be the deployment of a decision support system for maintenance management using an embedded analysis in the system itself, as seen in Figure 2.

A generic vehicle architecture (GVA) defines an open architecture over an information-centric design or a data-centric approach. The GVA has been standardized by NATO and is in the process of being standardized by its member states. An open architecture allows the interoperability of different devices from different developers. The NGVA standard is continuously developed by the Military Vetronics Association (MILVA) Working Group [20], which is an association of government agencies and industries that promote vehicle electronics (Vetronics) in military vehicles. The architecture integrates all mission systems, including data storage and processing, status monitoring of integrated systems, maintenance, and logistics in a system.

HUMS is a software platform that aims to integrate all information from installed systems to provide reports on the overall condition of the vehicle. The integration of native vehicle communication buses, such as control units in the Controller Area Network (CAN) bus and isolated sensors in legacy systems, is performed by defining data gateways for each network in a Generic Architecture.

The Military CAN (MilCAN A) specification [21] defines a deterministic protocol that can be applied to controller area network (CAN) technology as specified by ISO 11898 [22]. The MilCAN A specification is intended to support implementation in all areas of military vehicle application and is easily bridgeable to other CAN-based protocols such as SAE J1939 [23] and CAN open [24]. The physical layer of MILCAN uses a multidrop device topology in a single main bus. Each node may be linked to the main bus. MILCAN is directly bridgeable with the CAN bus (ISO 11898 standard [22]). It can support bitrates between 250 Kbps and 1 Mbps over a maximum 40 m bus length and up to 255 nodes. The military and industrial vehicle data frame uses an extended format identifier (ID) of 29 bits [25] according to the SAE J1939 data link protocol. The Logical Link Control layer in SAE J1939 describes this 29-bit identifier as part of the first 4 bytes of the frame. According to the Standard, the lower byte identifies up to 255 source addresses of the frame transmitting node. The second byte represents the message subtype as a page number (PGN), and the third byte identifies the function field. Each variable is associated with a “Suspect Parameter Number” (SPN), and MILCAN/SAE J1939 message selection is performed by means of a bit value of 25.

Parameters are generally coded using an 8-byte payload in the data frame. They are multiplexed according to the frame structure of the SAE J1939-71 application layer [26]. The standard allows the use of public and proprietary frames. Most of the public J1939 messages used are broadcasted at a fixed repetition rate which can vary from milliseconds to seconds, or they are requested messages. In general, diagnostic messages give trouble codes (DTC) for a suspect parameter together with a failure mode indicator (FMI) [27,28], using a request message with the desired SPN in a 3-byte datagram. Table 1 shows a list of common data frames in most vehicles with their related subsystems, according to identifiers and electronic source according to address. They are frequently broadcasted on the CAN bus network.

For data interpretation, the frame payload must be translated from hexadecimal to a human-readable format. A Fleet Management Systems (FMS) is a common interface that defines these supported parameters from major truck manufacturers. An FMS allows you to have manufacturer-independent applications to evaluate information and perform diagnostics. FMS enables a way or procedure for data interpretation for managing any or all aspects related to a fleet of vehicles operated by a company, government, or other organization as a common diagnostic framework. Although the Standard defines most of the well-known parameters, several manufacturers use proprietary coding for several parameters.

Modern tactical vehicles such as heavy trucks, wheeled all-road vehicles, and chain vehicles use CAN networks for internal system communications. Most of the platform sensors are directly attached to Electronic Control Units (ECUs) and communicate through the CAN network; however, the ECU itself has limited computing and data storage capabilities. They frequently only can store the minimum amount of information, like time use and system current system alarms. However, the CAN network and the overall J1939 protocol for data communications between vehicle subsystems are not considered suitable for data exchange with other mission systems. The exchanged data can be useful for driving, monitoring, and vehicle fault analysis. In addition, environmental sensors can provide situational awareness information. Therefore, we need to create an abstraction layer for the communication network intended to exchange information between systems. Ethernet-based virtualized network data communication is recommended in the generic vehicle architecture, as we will see below. In these premises, an on-board computer is suitable for registering data from sensors and vehicle networks, and shaping, transforming, analyzing, and distributing the data. All these tasks have been deployed in three main modules: data acquisition, condition analysis, and communication modules as shown in Figure 3.

An on-board computer for health and use monitoring was implemented using a Raspberry Pi-based Single Board Computer (SBC) with a reduced instruction system computer (RISC) processor for less consumption and a smaller factor. It has wireless, Bluetooth, and Ethernet connectivity. It uses a BCM2835 ARM RISC processor with quad core 1.4 GHz 64 bit and 1 Gbit SPDDR2 RAM. A General-Purpose Input Output port (GPIO) allows the addition of different sensors such as Inertial Measurement Units (IMUs), environment temperature sensors, a Global Positioning System (GPS/GNSS), cellular networks (3G/4G/LTE), etc. The prototype used a custom Debian operating system on a 64 Gbit SD card. To bring full connectivity to the vehicle buses, this board has a PiCAN2 CAN interface board [29].

As we can see in Figure 4, the sensor data published by the ECU are captured from the CAN bus or J1708 serial interface under the standard SAE J2012DA [30]. Several commercial data loggers use MDF4 binary files [31] to register CAN data in binary format files [32]. MDF4 files must be converted using proprietary modules or a Python converter module [33]. On the contrary, we used a Linux CAN module [34] to convert byte frames into plain text files. STANAG 4754 recommends using external gateways for legacy system communication; instead, information from different sources use internal interfaces and software gateways, pulling data into a virtual CAN, assuring source isolation and homogeneous data format. We obtain a simple way to transform ‘Socket CAN’ format files and a J1939 frame decoder (CAN tools module) which translates and converts the data to JavaScript Object Notation (JSON) or Comma Separated Value (CSV) files. Data selection for decoding is easily configured according to a CAN database file (DBC) [35] and CAN Python libraries [36]. From broadcasted functional data and using the above-mentioned decoding method and filtering data trouble code (DTC) according to the SAE J1939-73 frames, we extracted the diagnostic data for fault detection. The recorded data will have a two-fold use: human–machine interpretation of files through a human–machine interface, and data exchange with a vehicle model for mission systems, using a common vehicle data model. Each frame is timestamped, so it is easy to calculate each system use. The failure mode indicator and the status of the alarm lamp, in the same data frame, indicate the severity of the fault and the need for transmission. The development a prognostic data analyzer is pending for future fault prediction using off-board fault modelling and training.

The STANAG 4754 (NGVA) data model uses Local Area Networks (LAN) for its data communication infrastructure. Network segmentation in IP subnets allows one to securely manage sensor networks and data communication over the same physical network. According to STANAG, Vetronics data must use DDS middleware from Object Management Group [37] for interoperability with mission systems, which is currently in the integration process. The DDS protocol is a publish–subscribe message model, used to exchange data according to primitives and user-defined data types. A Vehicle Data Model provides a global data space for the exchange of information about vehicle configuration and useful data for the mission [38]. STANAG does not require the use of DDS communications, while ensuring no loss of reliability in the DDS network. Due to the high volume of registered functional data and the restricted communications, only a small amount of data can be exchanged using DDS for tactical communications. For data file exchange and downloading we have instead deployed a Web Representational State Transfer (REST) server. The server uses Hyper Text Transfer Protocol (HTTP) or secure (HTTPS) for human interaction. Alternatively, a simple storage server using the Amazon protocol (S3) [39] has been implemented as an embedded file server [40], which can be used for file replication on external servers and data visualization for technical purposes.

## 4. Results and Discussion

### 4.1. Test Context, Environment, and Data Sources

For testing purposes, commercial ruggedized computers with an ARM RISC processor (same as Raspberry PI) devices were selected for data acquisition, which were installed in Spanish wheeled tactical vehicles (Figure 5a). The device was connected to the diagnostic connector J1939 (Figure 5b), which provides permanent power while the battery is switched on. The data for testing was the real data registered [41] in those vehicles. To contextualize, the vehicles were used daily on an army base in normal conditions. The driver was aware of the installation of the acquisition device for functional analysis and the use of the vehicle. The data were grouped into parameter pages (PGN) according to SAE J1939-71 [26] and stored in a data lake.

### 4.2. Test Methodology and Metrics

Due to the lack of related works, we have defined three metrics for measuring the amount of data to classify system-related information. GTR measures the group time rate, SBR measures the storage byte rate in minutes, and TRR is the transmission rate in Kbps, τ is time and ψ is the number of frames.
(1)$$GTR\left(s\right)=\frac{\tau }{\psi }$$
(2)$$SBR\left(\frac{kByte}{\min}\right)=GTR\cdot 8\cdot \frac{60}{1000}$$
(3)$$TRR\left(\mathrm{Kbps}\right)=\frac{12\cdot 8\cdot \psi }{\tau }$$

Based on the metrics, a filtering method was implemented to classify each parameter group. Additionally, for fault detection, a Python decoder using CAN tools modules was implemented for diagnostic trouble code detection (DTC) and fault mode indication in the real-time condition analysis.

After classifying the received frames on the CAN bus according to SAE J1939-21 [42], at least eight emitting electronic units were identified when grouping frames by sender address, as we can see in Figure 6. Figure 7 shows the number of filtered packets, classified by parameter group, which is colored by the system emitter or electronic unit (ECU).

Additionally, using the CAN tools’ plot option allows one to visualize every time-related parameter for graphical identification of functional conditions during the normal use of the system (see Figure 8). We can distinguish idle conditions as an engine speed of 850 RPM, a stopped engine as 0 RPM, or running engine when speed is over 850 RPM. These conditions must be considered, according to the speed of the vehicle, for vehicle attitude evaluation. Using these rules, we can send the functional conditions for mission evaluation purposes.

Table 2 shows the measured bitrate and memory storage of different parameter group numbers that determine the amount of data to be stored and analyzed for prognostics.

The frames decoded and translated using a Python decoder from the CAN tools modules in J1939 can be visualized in JSON format in Figure 9.

### 4.3. Discussion

Unknown emitters generally use proprietary messages (proprietary PGN), but its variables are not useful in cross-platform algorithms. We can see discovered frames classified by emitters in Figure 6. Regarding the emitter, the engine and transmission messages occupied 95% of received frames, and diagnostic frames were only 0.06% of the total received frames. We have determined the type and value of the system data by translating the main frames for interpretation, especially diagnostic frames for fault condition detection. These frames can be used to exchange data according to the conditions and restrictions of military communications. We have proven that diagnostic frames in these vehicles are broadcasted by the engine ECU every second. Translating these data frames and using a database rule for the SPNs, we can discover current faults or the current system status. Unfortunately, we could not discover any failures or poor conditions.

The work presented in [8] considered the use of DDS for the exchange of sensor data in unmanned vehicles using a broadband wireless network. In contrast, our work considered vehicle engine data and the challenging use of tactical radios with restricted bandwidth according to the results in Table 2. The work presented in [4] based its function on a deployable system for sending a large amount of data. In contrast, we are trying to minimize data transmission for remote condition monitoring, using only active diagnostic trouble, and proposing embedding a future analyzer for each subsystem fault prediction. The proposed system can be interoperable, exchanging fault data in an open architecture for tactical vehicles using publish–subscribe services. Additionally, we propose to download raw data or translated data files on demand using wireless points of an embedded web file server.

## 5. Conclusions

An IoT architecture prototype was developed using a RISC processor-based SBC platform as a proof-of-concept. We used open-source software for data acquisition and fusion. Data translation allowed for explicit engine fault detection. Data translation was configured using a module user-configurable file as the fleet management system. This platform was tested using a commercial computer device. Engine parameters can be graphically analyzed using graphical CAN tools for normal use, expert analysis, and diagnostics. We could download data files for further analysis and training models, using a wireless access point and an embedded web file server, to avoid collapsing tactical radio channels. The system could be improved with other weak signals from vibration sensors for fault prediction, through the coupled-neurons method as proposed in [43]. In the next steps, the system will be interoperable in an open architecture using publish–subscribe services such as DDS, according to STANAG 4754. For a predictive maintenance approach, a future data analysis module for identified subsystems has been presented as the next development according to the received electronic control unit’s data. This work brings a system for local condition computing to the edge, avoiding continuous data transmission in real time for prognostics, as we have seen in previous works. In addition, an analysis framework must be developed to easily deploy different condition analysis modules for the defined subsystems described above. Finally, this proposed computing architecture has been accepted for factory installation in tactical wheeled vehicles developed in a Spanish factory for future deployment of the predictive logistic system in the Spanish Army.

## Figures and Tables

**Figure 1 sensors-23-05645-f001:**
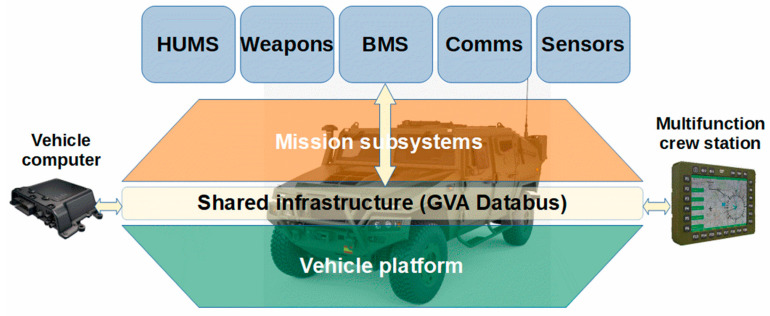
NGVA infrastructure using LOSA perspective.

**Figure 2 sensors-23-05645-f002:**
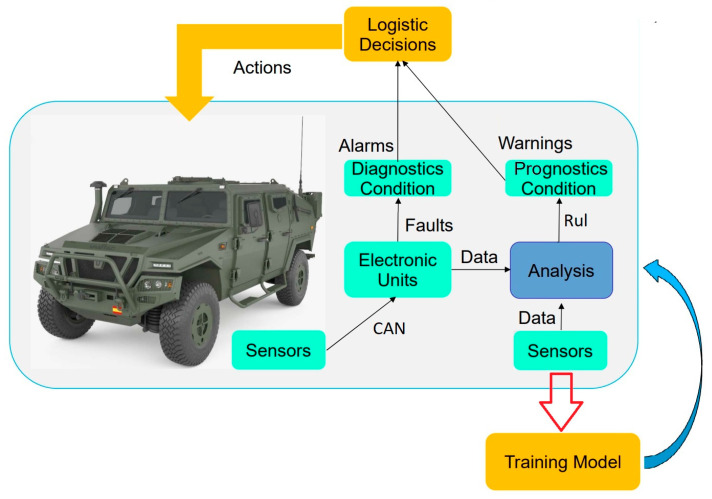
Condition and prognostic-based maintenance using on-board computing data. Yellow modules are off-board processes.

**Figure 3 sensors-23-05645-f003:**
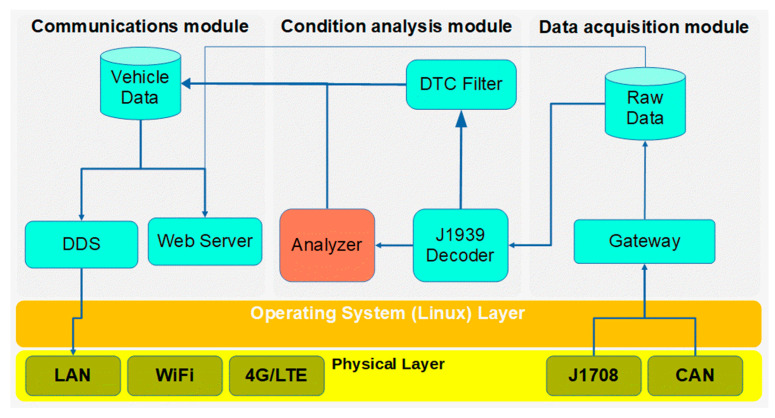
Proposed architecture for condition computing; red module not developed.

**Figure 4 sensors-23-05645-f004:**
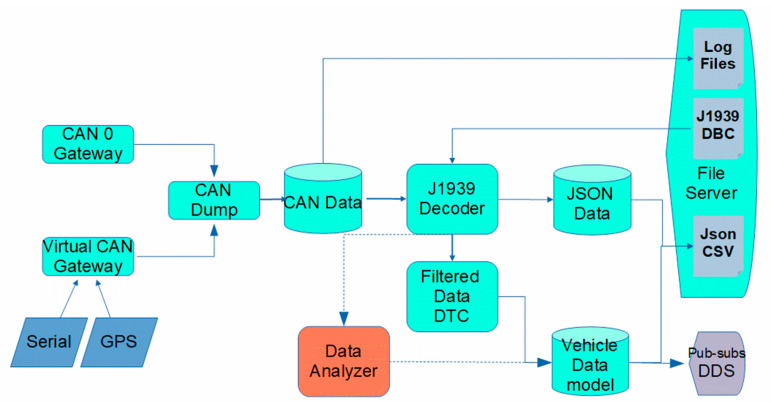
Implemented data flow and process; green modules were implemented and red module has not yet been developed.

**Figure 5 sensors-23-05645-f005:**
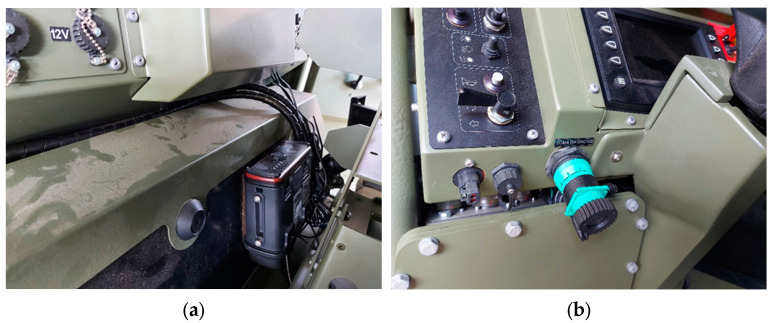
(**a**) Installed ruggedized computer. (**b**) Computer data and power installation.

**Figure 6 sensors-23-05645-f006:**
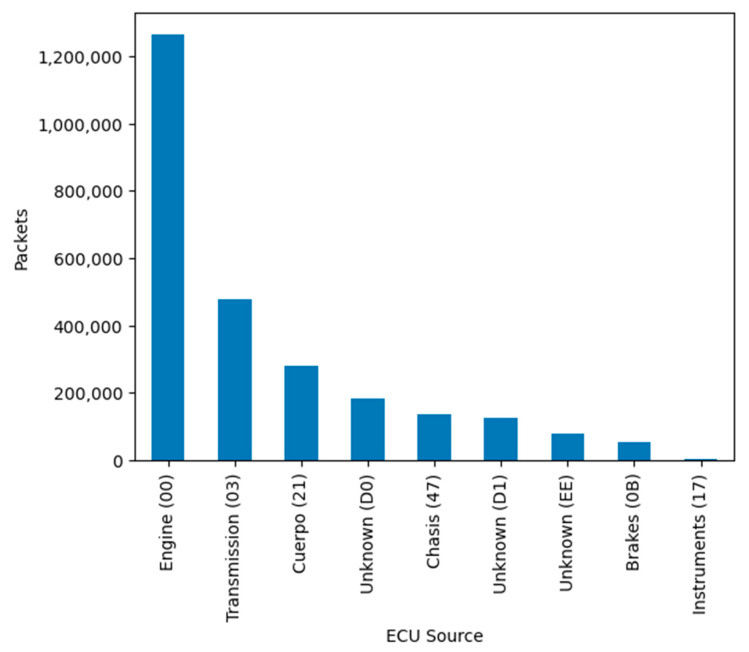
Grouped frames by Electronic Control Unit emitter.

**Figure 7 sensors-23-05645-f007:**
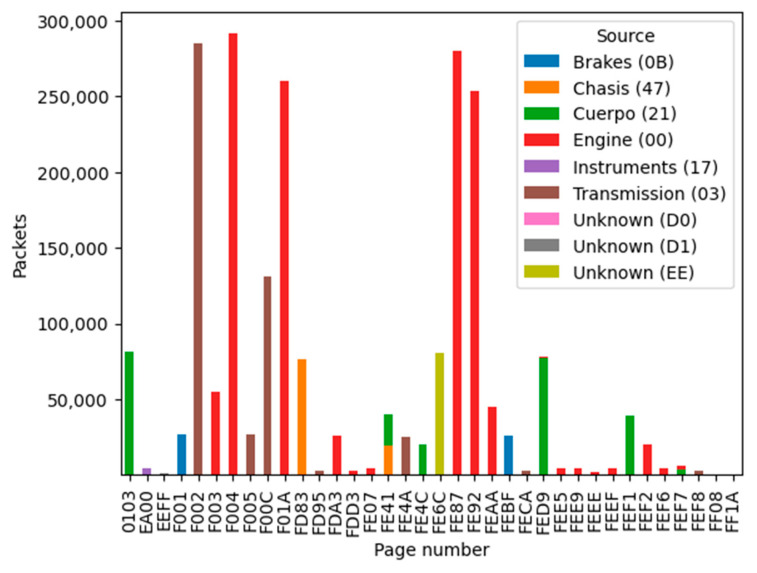
Grouped frames by page group number and ECU.

**Figure 8 sensors-23-05645-f008:**
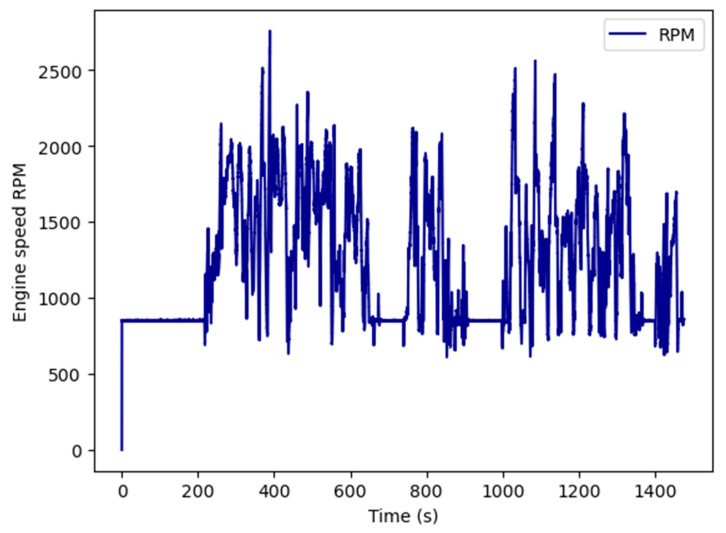
Graph of engine speed over time, using CAN Tools’ plot option.

**Figure 9 sensors-23-05645-f009:**
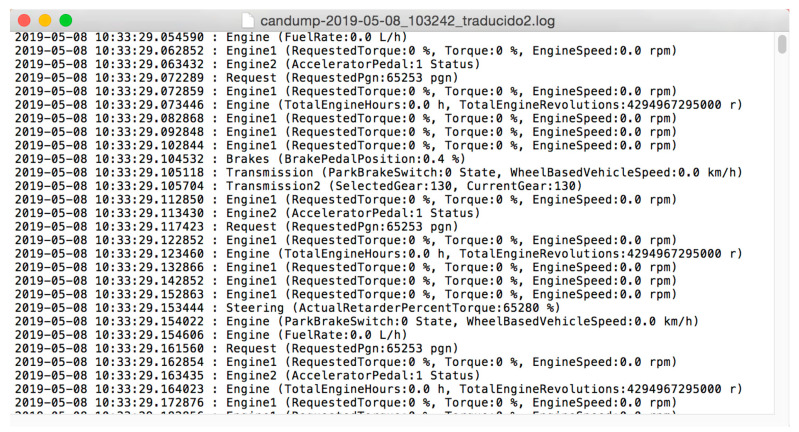
CAN dump frames with timestamp; decoded with J1939 CAN decoder module.

**Table 1 sensors-23-05645-t001:** Common SAE J1939-71 PGN codes with subsystems.

Code (Hex)	Frame Group	Frame Description	Subsystem	Source Address (Hex)
0000	TSC1	Torque/Speed Control	Engine	00
F000	ERC1	Retarder Controller	Retarder	0F
F001	EBC1	Brake Controller	Brakes	0B
F002	ETC1	Transmission Controller 1	Transmission	03
F003	EEC2	Engine Controller 2	Engine	00
F004	EEC1	Engine Controller 1	Engine	00
F005	ETC2	Transmission Controller 2	Transmission	03
F009	VDC2	Vehicle Dynamic Control 2	Body	21
FE6C	TCO1	Tachograph	Body	21
FE87	IT6	Ignition Timing 6	Engine	00
FEEE	ET1	Engine Temperature	Engine	00
FEEF	EFLP	Engine Fluid Level/Pressure	Engine	00
FEF1	CCVS1	Cruise Control/Vehicle Speed	Body	21
FEF2	LFE	Fuel Economy Liquid	Engine	00
FECA	DM1	Active Diagnostic Trouble Code	Engine	00

**Table 2 sensors-23-05645-t002:** Data transmission rate and memory usage (most relevant data).

Page (Hex)	Group (Code)	System	Frames (Number)	Frames (%)	GTR (s)	SBR (Kbyte)	TRR (Kbps)
0000	TSC1	Engine	31,734	1.39	0.047	10.212	1.92
F000	ERC1	Brake	15,189	0.66	0.1	4.8	0.96
F005	ETC1	Transmission	15,119	0.66	0.1	4.8	0.96
FEF7	ETC2	Transmission	151,880	6.69	0.01	48.0	9.6
F003	EEC2	Engine	30,377	1.33	0.05	9.6	1.92
F004	EEC1	Engine	151,886	6.69	0.01	48.0	9.6
FEEE	ET1	Engine	1519	0.06	1.0	0.48	0.096
FECA	DM1	Diagnostic ^1^	1519	0.06	1.0	0.48	0.096
FEF2	LFE	Engine	15,189	0.66	0.1	4.8	0.96

^1^ Diagnostic messages DM1 are sent from Engine ECU every second for active alarm notice.

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
