# Peer review of "Data Acquisition for Condition Monitoring in Tactical Vehicles: On-Board Computer Development"

_sensors, 2023, doi:10.3390/s23125645_

Round 1
Reviewer 1 Report
The work proposes data acquisition for condition monitoring in tactical vehicles. Although the authors have covered all the aspects broadly and explained them appropriately, the following points need to be considered for the possible publication of the manuscript.
1. The title should be "Data acquisition for condition monitoring in tactical vehicles: On-Board Computer Development." Modify the title accordingly.
2. Get the necessary permissions for Figure 1.
3. Restructure Figure 2. Add the overall methodology diagram, including data flows, and explain it in section 3.
4. Highlight the novelty of the proposed work. Also, compare the effectiveness of the proposed approach over the previous approach (through some significant parameters.)
5. Overall work is good.
Moderate English grammar and spell check is required.
Author Response
- The title has been modified.
- This is an own figure, created from the proposed architecture in LOSA. Figure title also has been accordingly changed because it had a mistake.
- I have rebuild figure 3 for better comprehension accordingly to the explained flow in section 3.
- The proposed work is part of a complex and ambitious work. Regarding the military conditions and the adoption of the model in the industry, novelty has been highlighted in introduction and conclussion sections. No other yet known works has proposed edge computing for tactical vehicle prognostics, avoiding continuous sending telemetry used for off-board analysis.
- Thanks by your review and recommendations.
Reviewer 2 Report
The authors studied the data acquisition for condition monitoring in tactical vehicles. It is meaningful for engineering application. However, some issues could be addressed as below.
1. In introduction, the authors could add some flowchart of condition monitoring in tactical vehicles from data acquisition to intelligent maintenance.
2. In section 3, the authors could add some weak feature extraction methods for designing health indicators for condition monitoring such as Harmonic-Gaussian double-well potential stochastic resonance with its application to enhance weak fault characteristics of machinery, Nonlinear Dynamics, 2023, 111(8): 7293-7307; Coupled neurons with multi-objective optimization benefit incipient fault identification of machinery, Chaos, Solitons and Fractals, 2021, 145: 110813.
3. The authors could illustrate the sensors’ types used to condition monitoring of tactical vehicles. How do the authors fuse these data?
Author Response
- A data flow from sensor to maintenance decision using on-board analysis is shown in figure 2. I have done some changes in figure, for better comprehension of the on-board system. Also, figure 4 show data process from sensor data to readable condition information to network publish.
- I plan to add vibration sensors for armoured vehicles in the future. I think interesting your proposed paper, and I am considering to add in conclusions as a future work.
- The whole sensors integrated in vehicle engine system by using CAN network, but some sensors could be attacheed to the system using other interfaces. The proposed system uses a simple software artifact for data fusion from different sources on a simple virtual CAN which is a virtual network gateway, avoiding direct interaction with physical networks and mixing interfaces. This has been briefly explained in section 3.
Reviewer 3 Report
This manuscript proposes an onboard computing platform for data registration, storage, and transformation. Overall, the work is clearly described. The references are relevant. The manuscript has the following issues that must be addressed.
1. The original technical contribution of the work is not clear. Figure 3 shows the proposed system architecture. Are the elements in Figure 3 and Figure 4 all existing technologies? What new original contributions are being proposed in the manuscript? Are there any new algorithmic principles or data analysis methods?
2. Line 303 says about a Python code. It is not clear what algorithm is used for diagnostics and real-time condition monitoring.
3. The experimental condition is vaguely defined in Section 3.2. It is not clear whether the vehicle was idling or in motion during the data collection.
4. Are the metrics presented in Table 2 good or bad? Are there any benchmarks? It would be nice to provide graphics and charts for the data in Table 2. The texts in Figure 9 are hard to read.
5. Equations 1-3 present the performance formulas of the proposed system. Is there a way of assessing the reliability of the proposed system?
Author Response
- Regarding the elements of the system in figures 3 and 4, it uses open source modules for data acquisition and tansformation, and standardized communication protocols. The work analyze captured engine data for future condition analysis and proposes to do on-board condition analysis according to the register data, instead of sending all data out of vehicle. The work is part of a more complex work as explained in introduction. We are studying algorithms for several engine condition monitoring analysis (analysis module) which is not part of this work.
- I have rewritten this sentence, because it had a mistake. Innitially I used a customized Python algorithm but at last, a CANTool decoder module was used for customized decoding of diagnostic frames.
- I have improved the explanation of test conditions for a better comprehension.
- Metrics in table 2 are good according to the expected metrics regarding standard SAE J1939 metrics. Data metric have been graphicaly represented grouped by system and by parameter group in figures 6 and 7, to see system grouped amount of data.
- Communication metrics show the quantity of data required for external system condition monitoring and the amount of data to be stored in order to see the requirements for the storage. These data show the reason of proposing edge computing for only alarms and warnings exchanging and the developing of on-board condition analysis.